# Self-Supervised Generalisation with Meta Auxiliary Learning

**Shikun Liu**     **Andrew J. Davison**     **Edward Johns**
Department of Computing, Imperial College London
{shikun.liu17, a.davison, e.johns}@imperial.ac.uk

## Abstract

Learning with auxiliary tasks can improve the ability of a primary task to generalise. However, this comes at the cost of manually labelling auxiliary data. We propose a new method which automatically learns appropriate labels for an auxiliary task, such that any supervised learning task can be improved without requiring access to any further data. The approach is to train two neural networks: a label-generation network to predict the auxiliary labels, and a multi-task network to train the primary task alongside the auxiliary task. The loss for the label-generation network incorporates the loss of the multi-task network, and so this interaction between the two networks can be seen as a form of meta learning with a double gradient. We show that our proposed method, Meta AuXiliary Learning (MAXL), outperforms single-task learning on 7 image datasets, without requiring any additional data. We also show that MAXL outperforms several other baselines for generating auxiliary labels, and is even competitive when compared with human-defined auxiliary labels. The self-supervised nature of our method leads to a promising new direction towards automated generalisation. Source code can be found at https://github.com/lorenmt/maxl.

## 1   Introduction

Auxiliary learning is a method to improve the ability of a primary task to generalise to unseen data, by training on additional auxiliary tasks alongside this primary task. The sharing of features across tasks results in additional relevant features being available, which otherwise would not have been learned from training only on the primary task. The broader support of these features, across new interpretations of input data, then allows for better generalisation of the primary task. Auxiliary learning is similar to multi-task learning [5], except that only the performance of the primary task is of importance, and the auxiliary tasks are included purely to assist the primary task.

We now rethink this generalisation by considering that not all auxiliary tasks are created equal. In supervised auxiliary learning [21, 33], auxiliary tasks can be manually chosen to complement the primary task. However, this requires both domain knowledge to choose the auxiliary tasks, and labelled data to train the auxiliary tasks. Unsupervised auxiliary learning [11, 36, 35, 16, 1] removes the need for labelled data, but at the expense of a limited set of auxiliary tasks which may not be beneficial for the primary task. By combining the merits of both supervised and unsupervised auxiliary learning, the ideal framework would be one with the flexibility to automatically determine the optimal auxiliary tasks, but without the need to manually label these auxiliary tasks.

In this paper, we propose to achieve such a framework with a simple and general meta-learning algorithm, which we call Meta AuXiliary Learning (MAXL). We first observe that in supervised learning, defining a task can equate to defining the labels for that task. Therefore, for a given primary task, an optimal auxiliary task is one which has optimal labels. The goal of MAXL is then to automatically discover these auxiliary labels using only the labels for the primary task.

The approach is to train two neural networks. First, a multi-task network, which trains the primary task and the auxiliary task, as in standard auxiliary learning. Second, a label-generation network, which learns the labels for the auxiliary task. The key idea behind MAXL is to then use the performance of the primary task, when trained alongside the auxiliary task in one iteration, to improve the auxiliary labels for the next iteration. This is achieved by defining the loss for the label-generation network as a function of the multi-task network's performance on primary task training data. In this way, the two networks are tightly coupled and can be trained end-to-end.

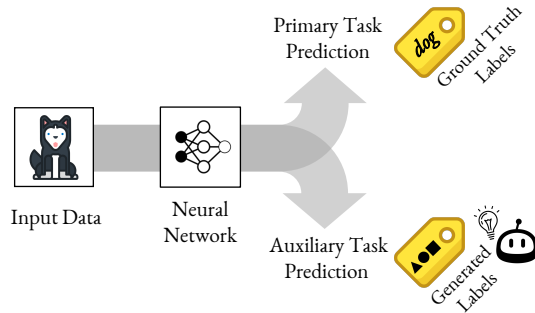

Figure 1: Illustration of MAXL framework. The primary task is trained with ground-truth labels, whereas the auxiliary task is trained with learned labels.

In our experiments on image classification, we show three key results. First, MAXL outperforms single-task learning across seven image datasets, even though both methods use the same amount of labelled data. Second, MAXL outperforms a number of baseline methods for creating auxiliary labels. Third, when manually-defined auxiliary labels exist, such as those from an image hierarchy, MAXL is at least as competitive, despite not actually using the manually-defined auxiliary labels. This last result shows that MAXL is able to remove the need for manual labelling of auxiliary tasks, which brings the advantages of auxiliary learning to new datasets previously not compatible with auxiliary learning, due to the lack of auxiliary labels.

## 2   Related Work

This work brings together ideas from a number of related areas of machine learning.

**Multi-task & Transfer Learning**     The aim of multi-task learning (MTL) is to achieve shared representations by simultaneously training a set of related learning tasks. In this case, the learned knowledge used to share across domains is encoded into the feature representations to improve performance of each individual task, since knowledge distilled from related tasks are interdependent. The success of deep neural networks has led to some recent methods advancing the multi-task architecture design, such as applying a linear combination of task-specific features [25, 8, 17]. [23] applied soft-attention modules as feature selectors, allowing learning of both task-shared and task-specific features in an end-to-end manner. Transfer learning is another common approach to improve generalisation, by incorporating knowledge learned from one or more related domains. Pre-training a model with a large-scale dataset such as ImageNet [7] has become a standard practise in many vision-based applications.

**Auxiliary Learning**     Whilst in multi-task learning the goal is high test accuracy across all tasks, auxiliary learning differs in that high test accuracy is only required for a single primary task, and the role of the auxiliary tasks is to assist in generalisation of this primary task. Applying related learning tasks is one straightforward approach to assist primary tasks. [33] applied auxiliary supervision with phoneme recognition at intermediate low-level representations to improve the performance of conversational speech recognition. [21] chose auxiliary tasks which can be obtained with low effort, such as global descriptions of a scene, to boost the performance for single scene depth estimation and semantic segmentation. By carefully choosing a pair of learning tasks, we may also perform auxiliary learning without ground truth labels, in an unsupervised manner. [16] introduced a method for improving agent learning in Atari games, by building unsupervised auxiliary tasks to predict the onset of immediate rewards from a short historical context. [11, 36] proposed image synthesis networks to perform unsupervised monocular depth estimation by predicting the relative pose of multiple cameras. [9] proposed to use cosine similarity as an adaptive task weighting to determine when a defined auxiliary task is useful. Differing from these works which require prior knowledge to manually define suitable auxiliary tasks, our proposed method requires no additional task knowledge, since it generates useful auxiliary knowledge in a purely unsupervised fashion. The most similar work to ours is [35], in which meta learning was used in auxiliary data selection. However, this still requires manually-labelled data from which these selections are made, whilst our method is able to generate auxiliary data from scratch.

**Meta Learning**    Meta learning (or learning to learn) aims to induce the learning algorithm itself. Early works in meta learning explored automatically learning update rules for neural models [4, 3, 29]. Recent approaches have focussed on learning optimisers for deep networks based on LSTMs [26] or synthetic gradients [2, 15]. Meta learning has also been studied for finding optimal hyper-parameters [20] and a good initialisation for few-shot learning [10]. [28] also investigated few shot learning via an external memory module. [34, 31] realised few shot learning in the instance space via a differentiable nearest-neighbour approach. Related to meta learning, our framework is designed to learn to generate useful auxiliary labels, which themselves are used in another learning procedure.

# 3    Meta Auxiliary Learning

We now introduce our method for automatically generating optimal labels for an auxiliary task, which we call Meta AuXiliary Learning (MAXL). In this paper, we only consider a single auxiliary task, although our method is general and could be modified to include several auxiliary tasks. We only focus on classification tasks for both the primary and auxiliary tasks, but the overall framework could also be extended to regression. As such, the auxiliary task is defined as a sub-class labelling problem, where each primary class is associated with a number of auxiliary classes, in a two-level hierarchy. For example, if manually-defined labels were used, a primary class could be "Dog", and one of the auxiliary classes could be "Labrador".

## 3.1    Problem Setup

The goal of MAXL is to generate labels for the auxiliary task which, when trained alongside a primary task, improve the performance of the primary task. To accomplish this, we train two networks: a *multi-task network*, which trains on the primary and auxiliary task in a standard multi-task learning setting, and a *label-generation network*, which generates the labels for the auxiliary task.

We denote the multi-task network as a function $f_{\theta_1}(x)$ with parameters $\theta_1$ which takes an input $x$, and the label-generation network as a function $g_{\theta_2}(x)$ with parameters $\theta_2$ which takes the same input $x$. Parameters $\theta_1$ are updated by losses of both the primary and auxiliary tasks, as is standard in auxiliary learning. However, $\theta_2$ is updated only by the performance of the primary task.

In the multi-task network, we apply a hard parameter sharing approach [27] in which we predict both the primary and auxiliary classes using the shared set of features $\theta_1$. At the final feature layer, $f_{\theta_1}(x)$, we then further apply task-specific layers to output the corresponding prediction for each task, using a SoftMax function. We denote the primary task predictions by $f_{\theta_1}^{\text{pri}}(x)$, and the auxiliary task predictions by $f_{\theta_1}^{\text{aux}}(x)$. And we denote the ground-truth primary task labels by $y^{\text{pri}}$, and the generated auxiliary task labels by $y^{\text{aux}}$.

We found during experiments that training benefited from assigning each primary class its own unique set of possible auxiliary classes, rather than sharing all auxiliary classes across all primary classes. In the label-generation network, we therefore define a hierarchical structure $\psi$ which determines the number of auxiliary classes for each primary class. At the output layer of the label-generation network, we then apply a masked SoftMax function to ensure that each output node represents an auxiliary class corresponding to only one primary class, as described further in Section 3.3. Given input data $x$, the label-generation network then takes in the hierarchy $\psi$ together with the ground-truth primary task label $y^{\text{pri}}$, and applies Mask SoftMax to predict the auxiliary labels, denoted by $y^{\text{aux}} = g_{\theta_2}^{\text{gen}}(x, y^{\text{pri}}, \psi)$. A visualisation of the overall MAXL framework is shown in Figure 2. Note that we allow soft assignment for the generated auxiliary labels, rather than one-hot encoding, which we found during experiments enables greater flexibility to obtain optimal auxiliary labels.

## 3.2    Model Objectives

The multi-task network is trained alongside the label-generation network, with two stages per epoch. In the first stage, the multi-task network is trained using primary task ground-truth labels, and the auxiliary labels from the label-generation network. In the second stage, the label-generation network is updated by computing its gradients with respect to the multi-task network's prediction accuracy on the primary task. We train both networks in an iterative manner until convergence.

In the first stage of each epoch, given target auxiliary labels as determined by the label-generation network, the multi-task network is trained to predict these labels for the auxiliary task, alongside the

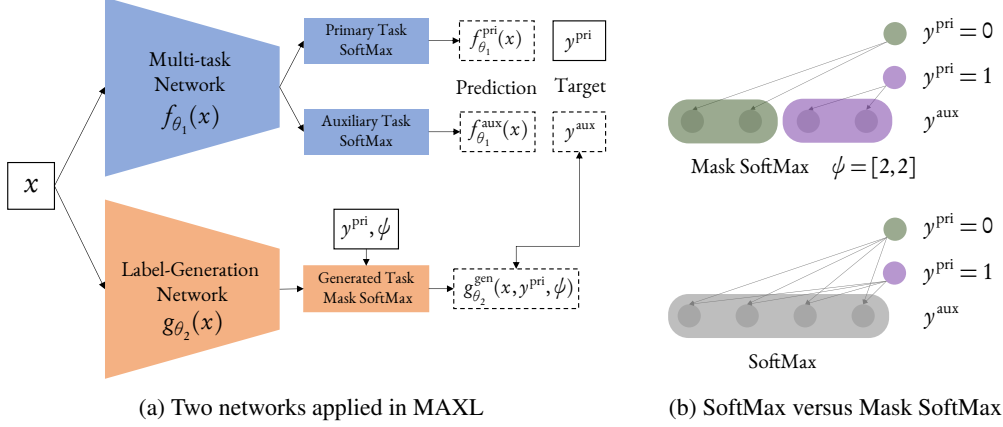

(a) Two networks applied in MAXL       (b) SoftMax versus Mask SoftMax

Figure 2: (a) Illustration of the two networks which make up MAXL. Dashed white boxes represent data generated by neural networks, solid white boxes represent given data, and coloured boxes represent functions. The double arrow represents equivalence. (b) Illustration of vanilla SoftMax and Mask SoftMax with 2 primary classes. Vanilla SoftMax outputs over all 4 auxiliary classes, whereas Mask SoftMax outputs over a hierarchical structure $\psi = [2, 2]$.

ground-truth labels for the primary task. For both the primary and auxiliary tasks, we apply the focal loss [22] with a focusing parameter $\gamma = 2$, defined as:

$$\mathcal{L}(\hat{y}, y) = -y(1 - \hat{y})^{\gamma} \log(\hat{y}), \tag{1}$$

where $\hat{y}$ is the predicted label and $y$ is the target label. The focal loss helps to focus on the incorrectly predicted labels, which we found improved performance during our experimental evaluation compared with the regular cross-entropy log loss.

To update parameters $\theta_1$ of the multi-task network, we define the multi-task objective as follows:

$$\underset{\theta_1}{\arg\min} \left( \mathcal{L}(f_{\theta_1}^{\mathrm{pri}}(x_{(i)}), y_{(i)}^{\mathrm{pri}}) + \mathcal{L}(f_{\theta_1}^{\mathrm{aux}}(x_{(i)}), y_{(i)}^{\mathrm{aux}}) \right) \tag{2}$$

where $(i)$ represents the $i^{th}$ batch from the training data, and $y_{(i)}^{\mathrm{aux}} = g_{\theta_2}^{\mathrm{gen}}(x_{(i)}, y_{(i)}^{\mathrm{pri}}, \psi)$ is generated by the label-generation network.

In the second stage of each epoch, the label-generation network is then updated by encouraging auxiliary labels to be chosen such that, if the multi-task network were to be trained using these auxiliary labels, the performance of the primary task would be maximised on this same training data. Leveraging the performance of the multi-task network to train the label-generation network can be considered as a form of meta learning. Therefore, to update parameters $\theta_2$ of the label-generation network, we define the meta objective as follows:

$$\underset{\theta_2}{\arg\min} \, \mathcal{L}(f_{\theta_1^+}^{\mathrm{pri}}(x_{(i)}), y_{(i)}^{\mathrm{pri}}) \,. \tag{3}$$

Here, $\theta_1^+$ represents the weights of the multi-task network after one gradient update using the multi-task loss defined in Equation 2:

$$\theta_1^+ = \theta_1 - \alpha \nabla_{\theta_1} \left( \mathcal{L}(f_{\theta_1}^{\mathrm{pri}}(x_{(i)}), y_{(i)}^{\mathrm{pri}}) + \mathcal{L}(f_{\theta_1}^{\mathrm{aux}}(x_{(i)}), y_{(i)}^{\mathrm{aux}}) \right), \tag{4}$$

where $\alpha$ is the learning rate.

The trick in this meta objective is that we perform a derivative over a derivative (a Hessian matrix) to update $\theta_2$, by using a retained computational graph of $\theta_1^+$ in order to compute derivatives with respect to $\theta_2$. This second derivative trick was also proposed in several other meta-learning frameworks such as [10] and [35].

However, we found that the generated auxiliary labels can easily collapse, such that the label-generation network always generates the same auxiliary label. This leaves parameters $\theta_2$ in a local

minimum without producing any extra useful knowledge. Therefore, to encourage the network to learn more complex and informative auxiliary tasks, we further apply an entropy loss $\mathcal{H}(y^{\text{aux}})$ as a regularisation term in the meta objective. A detailed explanation of the entropy loss and the collapsing label problem is given in Section 3.4. Finally, we update MAXL's label generation network by

$$\theta_2 \leftarrow \theta_2 - \beta \nabla_{\theta_2} \left( \mathcal{L}(f_{\theta_1^+}^{\text{pri}}(x_{(i)}), y_{(i)}^{\text{pri}}) + \lambda \mathcal{H}(y_{(i)}^{\text{aux}}) \right). \tag{5}$$

Overall, the entire MAXL algorithm is defined as follows:

---

**Algorithm 1:** The MAXL algorithm

---

**Initialise:** Network parameters: $\theta_1, \theta_2$; Hierarchical structure: $\psi$
**Initialise:** Learning rate: $\alpha, \beta$; Entropy weighting: $\lambda$
**while** *not converged* **do**
    **for** *each training iteration $i$* **do**
        *# fetch one batch of training data*
        $(x_{(i)}, y_{(i)}^{\text{pri}}) \in (x, y)$
        *# auxiliary-training step*
        Update: $\theta_1 \leftarrow \theta_1 - \alpha \nabla_{\theta_1} \left( \mathcal{L}(f_{\theta_1}^{\text{pri}}(x_{(i)}), y_{(i)}^{\text{pri}}) + \mathcal{L}(f_{\theta_1}^{\text{aux}}(x_{(i)}), g_{\theta_2}(x_{(i)}, y_{(i)}^{\text{pri}}, \psi)) \right)$
    **end**
    **for** *each training iteration $i$* **do**
        *# fetch one batch of training data*
        $(x_{(i)}, y_{(i)}^{\text{pri}}) \in (x, y)$
        *# retain training computational graph*
        Compute: $\theta_1^+ = \theta_1 - \alpha \nabla_{\theta_1} \left( \mathcal{L}(f_{\theta_1}^{\text{pri}}(x_{(i)}), y_{(i)}^{\text{pri}}) + \mathcal{L}(f_{\theta_1}^{\text{aux}}(x_{(i)}), g_{\theta_2}(x_{(i)}, y_{(i)}^{\text{pri}}, \psi)) \right)$
        *# meta-training step (second derivative trick)*
        Update: $\theta_2 \leftarrow \theta_2 - \beta \nabla_{\theta_2} \left( \mathcal{L}(f_{\theta_1^+}^{\text{pri}}(x_{(i)}), y_{(i)}^{\text{pri}}) + \lambda \mathcal{H}(y_{(i)}^{\text{aux}}) \right)$
    **end**
**end**

---

### 3.3 Mask SoftMax for Hierarchical Predictions

As previously discussed, we include a hierarchy $\psi$ which defines the number of auxiliary classes per primary class. To implement this, we designed a modified SoftMax function, which we call Mask SoftMax, to predict auxiliary labels only for certain auxiliary classes. This takes ground-truth primary task label $y$, and the hierarchy $\psi$, and creates a binary mask $M = \mathcal{B}(y, \psi)$. The mask is zero everywhere, except for ones across the set of auxiliary classes associated with $y$. For example, consider a primary task with 2 classes $y = 0, 1$, and a hierarchy of $\psi = [2, 2]$ as in Figure 2b. In this case, the binary masks are $M = [1, 1, 0, 0]$ for $y = 0$, and $[0, 0, 1, 1]$ for $y = 1$.

Applying this mask element-wise to the standard SoftMax function then allows the label-prediction network to assign auxiliary labels only to relevant auxiliary classes:

$$\text{SoftMax:} \quad p(\hat{y}_i) = \frac{\exp \hat{y}_i}{\sum_i \exp \hat{y}_i}, \qquad \text{Mask SoftMax:} \quad p(\hat{y}_i) = \frac{\exp M \odot \hat{y}_i}{\sum_i \exp M \odot \hat{y}_i}, \tag{6}$$

where $p(\hat{y}_i)$ represents the probability of the generated auxiliary label $\hat{y}$ over class $i$, and $\odot$ represents element-wise multiplication. Note that no domain knowledge is required to define the hierarchy, and MAXL performs well across a range of values for $\psi$ as shown in Section 4.2.

### 3.4 The Collapsing Class Problem

As previously discussed, we introduce an additional regularisation loss, which we call the entropy loss $\mathcal{H}(\hat{y}_{(i)})$. This encourages high entropy across the auxiliary class prediction space, which in turn encourages the label-prediction network to fully utilise all auxiliary classes. The entropy loss calculates the KL divergence between the predicted auxiliary label space $\hat{y}_{(i)}$, and a uniform

distribution $\mathcal{U}$, for each $i^{th}$ batch. This is equivalent to calculating the entropy of the predicted label space, and is defined as:

$$\mathcal{H}(\hat{y}_{(i)}) = \sum_{k=1}^{K} \hat{y}_{(i)}^k \log \hat{y}_{(i)}^k, \quad \hat{y}_{(i)}^k = \frac{1}{N} \sum_{n=1}^{N} \hat{y}_{(i)}^k[n]. \tag{7}$$

where $K$ is the total number of auxiliary classes, and $N$ is the training batch size.

## 4 Experiments

In this section, we present experimental results to evaluate MAXL with respect to several baselines and datasets on image classification.

### 4.1 Experimental Setup

**Datasets**  We evaluated on seven different datasets, with varying sizes and complexities. One of these, CIFAR-100 [18], contains a manually-defined 2-level hierarchical structure, which we expanded into a 4-level hierarchy by manually assigning data for the new levels, to create a hierarchy of {3, 10, 20, 100} classes . This hierarchy was then used for ground-truth auxiliary labels for the *Human* baseline (see below). For the other six datasets: MNIST [19], SVHN [12], CIFAR-10 [18], ImageNet [7], CINIC-10 [6] and UCF-101 [32], a hierarchy is either not available or difficult to access, and so no ground-truth auxiliary labels exist. All larger datasets were rescaled to resolution $[32 \times 32]$ to accelerate training.

**Baselines**  We compare MAXL to a number of baselines. First, we compare with *Single Task*, which trains only with the primary class label and does not employ auxiliary learning. This comparison was done to determine whether MAXL could improve classification performance without needing any extra labelled data. Then, we compare to three baselines for generating auxiliary labels: *Random*, *K-Means*, and *Human*, to evaluate the effectiveness of MAXL's meta-learning for label generation. *Random* assigns each training image to random auxiliary classes in a randomly generated (well-balanced) hierarchy. *K-Means* determines auxiliary labels via unsupervised clustering using K-Means [13], performed on the latent representation of an auto-encoder, with clustering updated after every training iteration. *Human* uses the human-defined hierarchy of CIFAR-100, where the auxiliary classes are at a lower (finer-grained) level hierarchy to the primary classes. Note that in order to compare these baselines to *Human*, they were only evaluated on CIFAR-100 because this is the only dataset containing a human-defined hierarchy (and hence ground-truth auxiliary labels).

### 4.2 Comparison to Single Task Learning

First, we compare MAXL to a single-task learning baseline, to determine whether MAXL can improve recognition accuracy without needing access to any additional data. To test the robustness of MAXL, we evaluate it on on 3 different networks: a simple 4-layer ConvNet, VGG-16 [30], and ResNet-32 [14]. We used hyper-parameter search for all networks and applied regularisation methods in order to achieve optimal performance . Since the power of MAXL lies in its ability to work without domain knowledge, we tested MAXL across a range of hierarchies $\psi$, to study if it is effective without needing to tune this hierarchy for each dataset. Here, the hierarchies are well balanced such that $\psi[i]$ is the same for all $i$ (for all primary classes).

| Datasets | Backbone | Single | MAXL, $\psi[i] =$ | | | |
|---|---|---|---|---|---|---|
| | | | 2 | 3 | 5 | 10 |
| MNIST | 4-layer ConvNet | $99.57 \pm 0.02$ | $99.56 \pm 0.04$ | $\mathbf{99.71 \pm 0.02}$ | $99.59 \pm 0.03$ | $99.57 \pm 0.02$ |
| SVHN | 4-layer ConvNet | $94.05 \pm 0.07$ | $94.39 \pm 0.08$ | $94.38 \pm 0.07$ | $\mathbf{94.59 \pm 0.12}$ | $94.41 \pm 0.09$ |
| CIFAR-10 | VGG-16 | $92.77 \pm 0.13$ | $93.27 \pm 0.09$ | $93.47 \pm 0.08$ | $\mathbf{93.49 \pm 0.05}$ | $93.10 \pm 0.08$ |
| ImageNet | VGG-16 | $46.67 \pm 0.12$ | $46.82 \pm 0.14$ | $46.97 \pm 0.10$ | $\mathbf{47.02 \pm 0.11}$ | $46.85 \pm 0.11$ |
| CINIC-10 | ResNet-32 | $85.12 \pm 0.08$ | $85.66 \pm 0.07$ | $85.72 \pm 0.07$ | $\mathbf{85.83 \pm 0.08}$ | $85.80 \pm 0.10$ |
| UCF-101 | ResNet-32 | $53.15 \pm 0.12$ | $54.19 \pm 0.18$ | $55.39 \pm 0.16$ | $\mathbf{54.70 \pm 0.12}$ | $54.32 \pm 0.18$ |

Table 1: Comparison of MAXL with single-task learning, across a range of hierarchies. We reported results from three individual runs, and the best performance for each dataset is marked with bold.

Table 1 shows the test accuracy of MAXL and single-task learning, with each accuracy averaged over three individual runs. We see that MAXL consistently outperforms single-task learning across all six datasets, despite both methods using exactly the same training data. We also see that MAXL outperforms single-task learning across almost all tested values of $\psi$, showing the robustness of our method without requiring domain knowledge or a manually-defined hierarchy.

### 4.3 Comparison to Auxiliary Label Generation Baselines

Next, we compare MAXL to a number of baseline methods for generating auxiliary labels, on CIFAR-100. Here, all the baselines were trained without any regularisation, to isolate the effect of auxiliary learning and test generalisation ability purely from auxiliary tasks. This dataset has a manually-defined hierarchy, which is used in *Human* for ground-truth auxiliary labels. However, MAXL, *Random*, and *K-Means* do not require any human knowledge or manually-defined hierarchy to generate auxiliary labels. Therefore, as in Section 4.2, a hierarchy $\psi$ is defined, assigning each primary class a set of auxiliary classes. We created well-balanced hierarchies by assigning an equal number of auxiliary classes per primary class. For cases where the hierarchy was unbalanced by one auxiliary class, we randomly chose which primary classes are assigned each number of auxiliary classes in $\psi$. We ran each experiment three times and averaged the results,

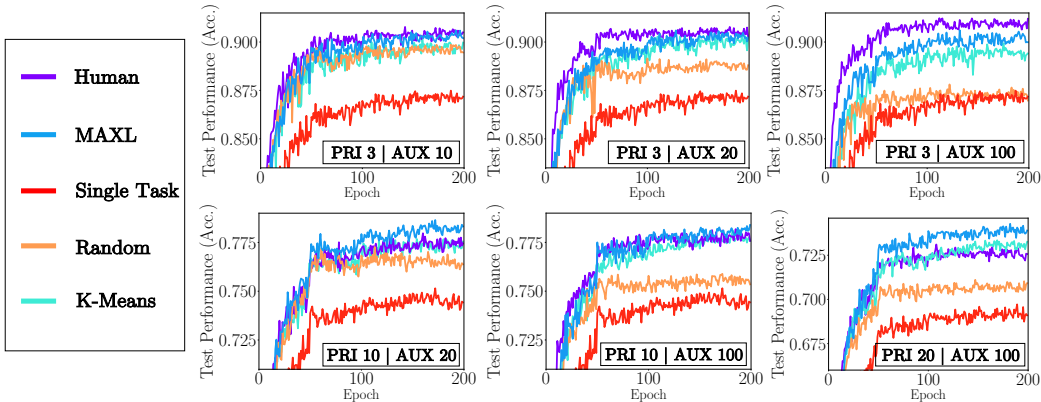

Figure 3: Learning curves for the CIFAR-100 test dataset, comparing MAXL with baseline methods for generating auxiliary labels. Our version of CIFAR-100 has a four-level hierarchy of {3, 10, 20, 100} classes per level, and we use this to create the hierarchy $\psi$ for auxiliary learning.

Test accuracy curves are presented in Figure 3, using all possible combinations of the numbers of primary classes and total auxiliary classes in CIFAR-100 (where the auxiliary classes are at a lower hierarchical level to the primary classes). We observe that MAXL outperforms *Single Task*, *Random*, and *K-Means*. Note that *K-Means* required significantly longer training time than MAXL due to the need to run clustering after each iteration. Also note that the superior performance of MAXL over these three baselines occurs despite all four methods using exactly the same data. Finally, we observe that MAXL performs similarly to *Human*, despite this baseline requiring manually-defined auxiliary labels for the entire training dataset. With performance of MAXL similar to that of a system using human-defined auxiliary labels, we see strong evidence that MAXL is able to learn to generalise effectively in a self-supervised manner.

### 4.4 Understanding the Utility of Auxiliary Labels

In [9], the cosine similarity between gradients produced by the auxiliary and primary losses was used to determine the task weighting in the overall loss function. We use this same idea to visualise the utility of a set of auxiliary labels for improving the performance of the primary task. Intuitively, a cosine similarity of -1 indicates that the auxiliary labels work against the primary task. A cosine similarity of 0 indicates that the auxiliary labels have no impact on the primary task. And a cosine similarity of 1 indicates that the auxiliary labels are learning the same features as the primary task, and so offer no useful information. Therefore, the cosine similarity for the gradient produced from optimal auxiliary labels should be between 0 and 1 to ensure that they assist the primary task.

In Figure 4, we show the cosine similarity measurements of gradients in the shared layers of the multi-task network, trained on 3 primary classes and 10, 20 and 100 total auxiliary classes from CIFAR-100. We observe that baseline methods *Human* and *Random*, with fixed auxiliary labels, reach their maximal similarity at an early stage during training, which then drops significantly afterwards. *K-Means* produces smooth auxiliary gradients throughout training, but its similarity depends on the number of auxiliary classes. In comparison, MAXL produces auxiliary gradients with high similarity throughout the entire training period, and consistently so across the number of auxiliary classes. Whilst we cannot say what the optimal cosine similarity should be, it is clear that MAXL's auxiliary labels affect primary task performance in a very different way to the other baselines.

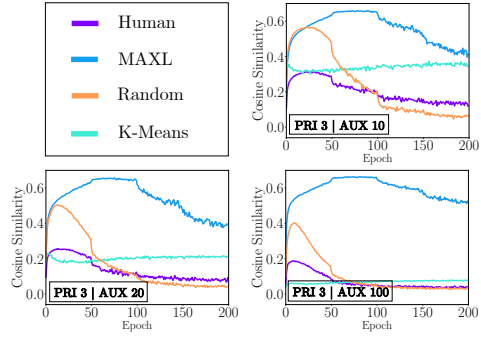

Figure 4: Cosine similarity measurement between the auxiliary loss gradient and primary loss gradient, on the shared representation in the multi-task network.

Due to MAXL's cosine similarity measurements being greater than zero across the entire training stage, a standard gradient update rule for shared feature space is then guaranteed to converge to a local minima given a small learning rate [9].

### 4.5 Visualisations of Generated Knowledge

In Figure 5, we visualise 2D embeddings of examples from the CIFAR-100 test dataset, on two different hierarchies. The visualisations are computed using t-SNE [24] on the final feature layer of the multi-task network, and compared across three methods: our MAXL method, the *Human* baseline, and the *Single Task* baseline.

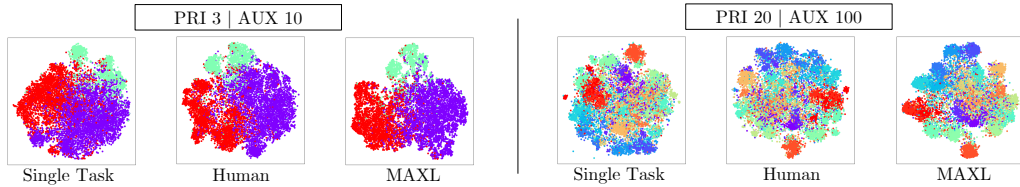

Figure 5: t-SNE visualisation of the learned final layer of the multi-task network, trained on CIFAR-100 with two different hierarchies. Colours represent the primary classes.

This visualisation shows the separability of primary classes after being trained with the multi-task network. Qualitatively, we see that both MAXL and *Human* show better separation of the primary classes than with *Single Task*, owing to the generalisation effect of the auxiliary learning. This again shows the effectiveness of MAXL whilst requiring no additional human knowledge.

We also show examples of images assigned to the same auxiliary class through MAXL's label-generation network. Figure 6 shows example images with the highest prediction probabilities for three random auxiliary classes from CIFAR-100, using the hierarchy of 20 primary classes and 100 total auxiliary classes (5 auxiliary classes per primary class), which showed the best performance of MAXL in Figure 3. In addition, we also present examples on MNIST, in which 3 auxiliary classes were used for each of the 10 primary classes.

To our initial surprise, only part of the generated auxiliary labels visualised in both dataset show human-understandable knowledge. We can observe that the auxiliary classes #1 and #2 of digit nine are clustered by the direction of the 'tail', and auxiliary classes #2 and #3 of digit seven are clustered by the distinction of the 'horizontal line'. But in most cases, there are no obvious similarities within each auxiliary class in terms of shape, colour, style, structure or semantic meaning. However, this makes more sense when we re-consider the role of the label-generation network, which is to assign auxiliary labels which assist the primary task, rather than grouping images in terms of semantic or visual similarity. The label-generation network would therefore be more effective if it were to group

Figure 6: Visualisation of 5 test examples with the highest prediction probability, for each of 3 randomly selected auxiliary classes, for different primary classes. We present the visualisation for CIFAR-100 (top) when trained with 20 primary classes and 5 auxiliary classes per primary class, and for MNIST (bottom) when trained with 10 primary classes and 3 auxiliary classes per primary class.

images in terms of a shared aspect of reasoning which the primary task is currently struggling to learn, which may not be human intepretable.

Furthermore, we discovered that the generated auxiliary knowledge is not deterministic, since the top predicted candidates are different when we re-train the network from scratch. We therefore speculate that using a human-defined hierarchy is just one out of a potentially infinite number of local optima, and on each run of training, the label-generation network produces another of these local optima.

## 5   Conclusion & Future Work

In this paper, we have presented Meta AuXiliary Learning (MAXL) for generating optimal auxiliary labels which, when trained alongside a primary task in a multi-task setup, improve the performance of the primary task. Rather than employing domain knowledge and human-defined auxiliary tasks as is typically required, MAXL is self-supervised and, combined with its general nature, has the potential to automate the process of generalisation to new levels.

Our evaluation on multiple datasets has shown the performance of MAXL in an image classification setup, where the auxiliary task is to predict sub-class, hierarchical labels for an image. We have shown that MAXL significantly outperforms other baselines for generating auxiliary labels, and is competitive even when human-defined knowledge is used to manually construct the auxiliary labels.

The general nature of MAXL also opens up questions about how self-supervised auxiliary learning may be used to learn generic auxiliary tasks, beyond sub-class image classification. During our experiments, we also ran preliminary experiments on predicting arbitrary vectors such that the auxiliary task becomes a regression, but results so far have been inconclusive. However, the ability of MAXL to potentially learn flexible auxiliary tasks which can automatically be tuned for the primary task, now offers an exciting direction towards automated generalisation across a wide range of more complex tasks.

## Acknowledgements

We would like to thank Michael Bloesch, Fabian Falck, and Stephen James, for insightful discussions.

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
