[Supplementary Material · MAXL_NIPS_supp.pdf]

# Self-Supervised Generalisation with Meta Auxiliary Learning (Supplementary Material)

## 1 4-level CIFAR-100 Dataset

| 3 Class | 10 Class | 20 Class | 100 Class |
|---------|----------|----------|-----------|
| animals | large animals | reptiles | crocodile, dinosaur, lizard, snake, turtle |
| | | large carnivores | bear, leopard, lion, tiger, wolf |
| | | large omnivores and herbivores | camel, cattle, chimpanzee, elephant, kangaroo |
| | medium animals | aquatic mammals | beaver, dolphin, otter, seal, whale |
| | | medium-sized mammals | fox, porcupine, possum, raccoon, skunk |
| | small animals | small mammals | hamster, mouse, rabbit, shrew, squirrel |
| | | fish | aquarium fish, flatfish, ray, shark, trout |
| | invertebrates | insects | bee, beetle, butterfly, caterpillar, cockroach |
| | | non-insect invertebrates | crab, lobster, snail, spider, worm |
| | people | people | baby, boy, girl, man, woman |
| vegetations | vegetations | flowers | orchids, poppies, roses, sunflowers, tulips |
| | | fruit and vegetables | apples, mushrooms, oranges, pears, peppers |
| | | trees | maple, oak, palm, pine, willow |
| objects and scenes | household objects | food containers | bottles, bowls, cans, cups, plates |
| | | household electrical devices | clock, keyboard, lamp, telephone, television |
| | | household furniture | bed, chair, couch, table, wardrobe |
| | construction | large man-made outdoor things | bridge, castle, house, road, skyscraper |
| | natural scenes | large natural outdoor scenes | cloud, forest, mountain, plain, sea |
| | vehicles | vehicles 1 | bicycle, bus, motorcycle, pickup truck, train |
| | | vehicles 2 | lawn-mower, rocket, streetcar, tank, tractor |

Table 1: CIFAR-100 dataset in 4-level hierarchy.

## 2 Training Strategies

For MAXL's multi-task network, we applied SGD with a learning rate of $0.01$ and we dropped the learning rate by half for every 50 epochs with a total of 200 epochs in 4-level CIFAR-100 dataset; we applied SGD with a learning rate of $0.1$ with momentum $0.9$ and weight decay of $5 \cdot 10^{-4}$ for the other 6 datasets and we used cosine annealing schedule to optimise the network until convergence.

For MAXL's label-generation network, we found that a smaller learning rate of $10^{-3}$ was necessary to help prevent the class collapsing problem, and we further applied a weight decay of $5 \cdot 10^{-4}$ in all evaluated datasets. We chose the weighting of the entropy regularisation loss term to be $\lambda = 0.2$ based on empirical performance.

# 3  Cosine Similarity on CIFAR-100 Dataset

Figure 1: Cosine similarity measurement between the auxiliary loss gradient and primary loss gradient on the shared representation in the multi-task network.

# 4  Further Analysis on the Collapsing Class Problem

In Table 2, we show results on CIFAR-100 with (left) and without (right) entropy loss for $\lambda = 0.2$ and 0 respectively, for all hierarchy structures. On the rightmost column, we show the test accuracy. On the second right column, we show the percentage of auxiliary labels which are actually utilised (assigned to by the label-generation network). We see that MAXL with entropy loss utilises the entire auxiliary space, and improves performance compared to using no entropy loss, because in this case, the label space is not fully utilised.

| PRI | AUX | Label % | Accuracy |
|-----|-----|---------|----------|
| 3 | 10 | 1.00 \| 1.00 | 90.50 \| 90.26 |
| 3 | 20 | 1.00 \| 0.65 | 90.65 \| 90.39 |
| 3 | 100 | 1.00 \| 0.35 | 90.66 \| 90.22 |
| 10 | 20 | 1.00 \| 1.00 | 78.40 \| 77.73 |
| 10 | 100 | 1.00 \| 0.57 | 78.46 \| 78.20 |
| 20 | 100 | 1.00 \| 0.61 | 74.27 \| 73.97 |

Table 2: Comparison of accuracies of 4-level CIFAR-100 dataset with and without entropy loss.

# 5  Negative Results

As well as those described in the paper, we explored a range of ideas for implementing MAXL which were not successful. We report these below in order to assist with guiding future work.

- We found that a standard cross-entropy loss leads to worse performance than the focal loss.

- We experimented with MAXL without Mask SoftMax, and it achieved a similar performance to single-task learning.

- We experimented with MAXL producing an auxiliary latent vector to be used for regression, rather than auxiliary labels to be used for classification, and it achieved similar performance to single-task learning.

- We tried updating the multi-task and label-generation networks in the same iteration, but we found that it led to worse performance than training each network independently for multiple iterations per epoch.

- We evaluated MAXL on semantic segmentation tasks, but we found that it only had marginal benefit.

- We tried a number of different designs for MAXL's label-generation network, but they had minimal effect on the final performance.

- We tried updating the label-generation network based on the multi-task networks performance on unseen validation data, but this was not as beneficial as updating based on the performance on the same training data.