[Reviews · NeurIPS 2019]

Reviewer 1



This paper addresses an important problem in a unique way. Self-supervision is a promising avenue of research, but currently relies on significant domain knowledge. The authors propose to overcome this with model agnostic meta learning. The experiments are fairly comprehensive and the exposition is clear and well-cited. Despite the originality of the work, the experiments do not currently make a very strong case for its significance. The comparisons in Table 1 show consistent but modest improvements in accuracy. While the improvements are greater than the run-to-run variation with different random seeds, it is unclear how such an improvement compares to variations in performance by modifying standard data augmentation and/or regularization. This makes it less clear how general the results would be. Similarly, the results on Cifar100 are suggestive but not very convincing. MAXL clearly performs significantly better than single task training and the random baseline, but the relative advantage over the k-means baseline seems to be at most 1% relative. It is difficult to tell by the graph presentation, and it's not clear what additional information is provided by the time series of the graph. Finally, it's nice that visualizations are presented, but the analysis of the visualizations is somewhat lacking. CNNs are notorious for modeling less salient features such as texture and global illuminance. Further quantitative or qualitative studies, such as salience / gradient visualization of the learned features could help illustrate the common characteristics.

Reviewer 2



Main Ideas The high level motivation is to combine the strengths of supervised and unsupervised methods for auxiliary task learning. The authors present a meta-learning algorithm that automatically determines labels of auxiliary tasks without manual labels. They study this method in the context of classification. Relation to Prior Work This is a straightforward application of gradient-based meta-learning. The formulation of tailoring the learning of the label-generator to the learning progress of the multi-task learner is an elegant formualtion of an iterative optimization procedure. Quality - strengths: the authors conducted a thorough analysis comparing MAXL with several baselines. - weakness: It would strengthen the paper to show an experiment that analyzes the weighting coefficient lambda on the entropy term. The authors state the collapsing class problem, but do not show an experiment highlighting why the problem is important. The number of auxiliary classes per primary class seems to be a hyperparameter; it would be informative if the authors could provide an analysis for how to choose this hyperparameter, as according to Figure 3 the choice of hyperparameter has a non-trivial effect on generalization performance. Clarity - strengths: the paper is very well written and motivated Originality - strengths: the proposed method seems to be novel Significance - strengths: MAXL can be in principled be applied to any classification task as long as the number (but not the identity) of auxiliary tasks is pre-defined. - weakness: While MAXL provides an improvement over single task learning as shown in Table 1, the improvement seems marginal. It would be informative for the authors to include a discussion for why MAXL could not improve generalization performance beyond one percentage point in all of the classification tasks.

Reviewer 3



Though I think this paper proposed a very interesting approach to automating the design of auxiliary tasks. I am disappointed by its practical value on the image classification tasks evaluated. According to Table 1, the method outperformed the standard single-task learning baseline by a very small margin (less than 1%) on all seven datasets. Why didn’t we see larger performance gains using the proposed approach? I’d hope to hear the authors’ hypothesis. Also, with what kind of datasets/models, the proposed method for generating auxiliary tasks is the most effective? I am also curious about the scalability of this approach with more advanced architectures and larger-datasets. In the experiments, larger images are rescaled to [32x32] low resolution. Could the authors comment on the reason of this design choice? Are there any technical constraints that limit the model in training with larger images? Would this model be effective in more realistic image classification setups, say training the state-of-the-art ResNet-101 for the ImageNet challenge? Another baseline that I’d like to see is to use variational encoders (e.g. beta-VAE) to learn latent discrete representations from the data and use the learned discrete code as labels for auxiliary learning. This could possibly be a stronger baseline than the k-means baseline. Fig. 4: what’s the intuition why the cosine similarity between the gradients of the primary task and the ones of the auxiliary task always possible? Would it be possible that the auxiliary tasks can generate gradients in the opposite direction to the gradient of the primary task so that it pulls it out of a local minimum? Fig: 2: is there any parameter sharing between the multi-task network and the label generation network? I appreciate the provided source code and the reported negative results in the supplementary materials.

[Author Response · NeurIPS 2019]

Thank you to all three reviewers for your positive and constructive feedback. Before addressing the specific points raised, we would first like to emphasise that the primary objective of this work was not to achieve a new state-of-the-art benchmark by incorporating all the latest network architectures, and carrying out extensive hyperparameter search. Instead, our contribution is primarily conceptual and explorative: we show that, without requiring any additional data, auxiliary task labels can be automatically generated to improve the performance of a primary task. This is a very different way of thinking about machine learning to existing approaches to supervised learning, and we believe that this sets the scene for a new direction from which significant future work can emerge.

– **Marginal Performance Improvement (to R1, R2, R3)** The major concern among all reviewers is the marginal improvement of MAXL in Table 1, even though reviewers acknowledge that the improvements are robust across datasets and network architectures, and are statistically significant. We accept this. However, improving performance when no additional data is available, is notoriously difficult. For example, improvements due to data augmentation (Mixup [Zhang et al. 2018], CutOut [DeVries et al. 2017]) and gradient manipulation (Shake-Shake regularisation [Gastaldi 2017]), all show $< 2\%$ improvement on CIFAR-10. As such, we are achieving similar improvements to various state-of-the-art regularisation techniques, without even including such regularisation. Furthermore, whilst results in Table 1 are compared to scores presented elsewhere using methods which employ significant regularisation (line 221-222), results in Figure 3 are a purer comparison, where none of the implementations use any regularisation (line 233-234). Here, we see a much more dramatic improvement of MAXL over single-task learning (around 4–6%). This shows that the performance increase due to MAXL alone, when regularisation is not an influence, is actually much greater than Table 1 may at first suggest.

– **More Baselines (to R1, R3)** Together with the baselines presented in this paper, we did in fact experiment with discrete VAE using gumbel softmax [Jang et al, 2017] and Prototypical Networks [30]. However, both of those methods performed only marginally above baseline *Random*, and below the *K-means* baseline. Considering K-means is a more popular unsupervised clustering approach, we decided to present only K-means in this paper to avoid overcrowding with similar baselines. However, we will add a short discussion of these other baselines experiments in the camera-ready paper.

– **Weighting Coefficient / Collapsing Class Problem (to R2)**

We agree with R2 that additional experiments on various weight coefficients on entropy loss can lead to a better understanding of the collapsing class problem. In this new table, we show results on CIFAR-100 with (left) and without (right) entropy loss for $\lambda = 0.2$ and 0 respectively, for all hierarchy structures. On the rightmost column, we show the test accuracy. On the second right column, we show the percentage of auxiliary labels which are actually utilised (assigned to by the label-generation network). We see that

| PRI | AUX | Label % | Accuracy |
|---|---|---|---|
| 3 | 10 | 1.00 \| 1.00 | 90.50 \| 90.26 |
| 3 | 20 | 1.00 \| 0.65 | 90.65 \| 90.39 |
| 3 | 100 | 1.00 \| 0.35 | 90.66 \| 90.22 |
| 10 | 20 | 1.00 \| 1.00 | 78.40 \| 77.73 |
| 10 | 100 | 1.00 \| 0.57 | 78.46 \| 78.20 |
| 20 | 100 | 1.00 \| 0.61 | 74.27 \| 73.97 |

MAXL with entropy loss utilises the entire auxiliary space, and improves performance compared to using no entropy loss, because in this case, the label space is not fully utilised. We will provide a detailed explanation in the camera-ready paper.

– **Auxiliary Hierarchy (to R2)** In Table 1, we show an ablative study on the number of auxiliary classes $\psi$. In all cases, MAXL improved performance over single-task learning, showing that MAXL is robust to choice of this hyperparameter. Our preliminary findings do, however, show that there is no one value for $\psi$ which consistently outperforms all others. As such, automatically determining the optimal hierarchy from training data alone presents intriguing future work in response to the promising results in this first paper.

– **Parameter Sharing (to R3)** There is no parameter sharing between multi-task network and label-generation network. We wanted to have a fair comparison between MAXL and other baseline methods, and thus we trained the primary task using the same network, for all methods.

– **Cosine Similarity (to R3)** If auxiliary task gradients have opposite direction to the primary task gradients, MAXL will not be guaranteed to converge to a local minimum according to Proposition 1 in [9]. The success of MAXL lies in the fact that the generated auxiliary labels provide a similar but different gradient compared to the primary task.

– **Image Resolution (to R3)** Since ImageNet is a very large dataset, we re-scaled the images to a smaller resolution due to the limited hardware available in our lab, in order to generate the large number of experimental results presented. Promising future work would be to investigate using a first-order approximation to the second-order gradient, to speed up MAXL for use with limited hardware on higher-resolution images.

[Meta-Review · NeurIPS 2019]

This paper addresses a method to improve a primary task by jointly learning auxiliary tasks. A unique contribution in this paper is in the label generation network which is trained in a self-supervised manner to generate labels for the auxiliary task. A way to jointly train both multi-task network and label-generation network is similar to MAML. In that sense, the method is referred to as meta auxiliary learning, but such name could be misleading. All of reviewers agree that the paper has an interesting idea. On the other hand, there are concerns on experiments where only marginal improvement over single task learning is shown. During the discussion, one reviewer raised his/her score, but they still have concerns that improvements are marginal in image classification tasks.